# Exercise and Interorgan Communication: Short-Term Exercise Training Blunts Differences in Consecutive Daily Urine ^1^H-NMR Metabolomic Signatures between Physically Active and Inactive Individuals

**DOI:** 10.3390/metabo12060473

**Published:** 2022-05-24

**Authors:** Leon Deutsch, Alexandros Sotiridis, Boštjan Murovec, Janez Plavec, Igor Mekjavic, Tadej Debevec, Blaž Stres

**Affiliations:** 1Biotechnical Faculty, Department of Animal Science, University of Ljubljana, SI-1000 Ljubljana, Slovenia; leon.deutsch@bf.uni-lj.si; 2Section of Sport Medicine and Biology of Exercise, School of Physical Education and Sport Science, National and Kapodistrian University of Athens, 17237 Athens, Greece; asotiridis@phed.uoa.gr; 3Department of Automation, Biocybernetics and Robotics, Jožef Stefan Institute, SI-1000 Ljubljana, Slovenia; igor.mekjavic@ijs.si (I.M.); tadej.debevec@fsp.uni-lj.si (T.D.); 4Faculty of Electrical Engineering, University of Ljubljana, Jamova 2, SI-1000 Ljubljana, Slovenia; bostjan.murovec@fe.uni-lj.si; 5NMR Center, National Institute of Chemistry, SI-1000 Ljubljana, Slovenia; janez.plavec@ki.si; 6Faculty of Sports, University of Ljubljana, SI-1000 Ljubljana, Slovenia; 7Faculty of Civil and Geodetic Engineering, Institute of Sanitary Engineering, University of Ljubljana, SI-1000 Ljubljana, Slovenia

**Keywords:** exercise, trained, untrained, ^1^H-NMR metabolomics, human metabolome, JADBio, biomarkers

## Abstract

Physical inactivity is a worldwide health problem, an important risk for global mortality and is associated with chronic noncommunicable diseases. The aim of this study was to explore the differences in systemic urine ^1^H-NMR metabolomes between physically active and inactive healthy young males enrolled in the X-Adapt project in response to controlled exercise (before and after the 3-day exercise testing and 10-day training protocol) in normoxic (21% O_2_), normobaric (~1000 hPa) and normal-temperature (23 °C) conditions at 1 h of 50% maximal pedaling power output (W_peak_) per day. Interrogation of the exercise database established from past X-Adapt results showed that significant multivariate differences existed in physiological traits between trained and untrained groups before and after training sessions and were mirrored in significant differences in urine pH, salinity, total dissolved solids and conductivity. Cholate, tartrate, cadaverine, lysine and N6-acetyllisine were the most important metabolites distinguishing trained and untrained groups. The relatively little effort of 1 h 50% W_peak_ per day invested by the untrained effectively modified their resting urine metabolome into one indistinguishable from the trained group, which hence provides a good basis for the planning of future recommendations for health maintenance in adults, irrespective of the starting fitness value. Finally, the 3-day sessions of morning urine samples represent a good candidate biological matrix for future delineations of active and inactive lifestyles detecting differences unobservable by single-day sampling due to day-to-day variability.

## 1. Introduction

Physical inactivity is a worldwide health problem ranking as the fourth most important risk for global mortality [1]. The efforts undertaken by the World Health Organization (WHO) to minimize the time spent sedentary [2] are directed at decreasing the risks for more than twenty chronic noncommunicable diseases (e.g., coronary heart disease, stroke, type 2 diabetes, obesity, metabolic syndrome, glucose insensitivity) next to mental health and neurological problems such as depression and dementia [1]. As physical activity in the form of various types of exercise promotes wellbeing and increased quality of life, understanding the biological mechanisms through which it impacts is of central importance. Although genetics, lifestyle and environment are likely the most important parameters, their relative contributions and interactions are not well-understood.

Exercise-related stress alters the chemical steady state of the internal biochemical environment. The net result is modifications in the rate of production and consumption of various metabolites within biochemical network affecting the systemic levels of metabolites relative to the exercise intensity, muscle damage or the extent of the exercise as part of the lifelong history [3,4,5].

In respect to the progressively emerging picture of metabolic states characteristic of various noncommunicable diseases, a number of metabolomic studies have clearly shown that physical activity results in modifications of hundreds of metabolites associated with fatty-acid mobilization, lipolysis and metabolism, the TCA cycle, glycolysis, amino-acid metabolism, carnitine, purine and cholesterol metabolism and insulin sensitivity [1]. Based on these results, it has become obvious that while the metabolomics patterns may differ slightly between groups, it was the overall volume of exercise acting as the most important driver of the metabolomics makeup [6,7,8,9], even irrespective of hypoxia [4]. This points to the multifactorial dose–response relationship between activity (intensity, frequency, time frame (exposure measured in hours, days, weeks, years, lifelong)) and metabolomic signatures [1,4].

Metabolomics has become a technology-driven discipline focusing on improved high-throughput and large-scale data collection, analysis and interpretation. Metabolomes were characterized utilizing proton nuclear magnetic resonance (^1^H-NMR) in minimally user-invasive biomaterial—the first morning urine. The approach of ^1^H-NMR was utilized in this study as it is nondestructive, quantitative, cost-effective, reproducible and requires no sample derivatization [10]. Although the approach captures a modest number of metabolites (*n* > 350 in our past studies [4,5,11]) it enables identification of unknown novel compounds in complex biological matrices such as serum, saliva, urine or feces [4,10] and has been frequently utilized (>40% of studies) in observational and experimental studies next to short-term (<1 week) or long-term (>1 week) interventions [1]. The use of consecutive three-day urine ^1^H-NMR data points was first tested recently in the form of a 3-day sliding window within the PlanHab project and showed promising results for delineation of systemic differences between groups [3,4,5]. 

The aim of this study was to explore the differences in systemic urine ^1^H-NMR metabolomic signatures between groups of physically active and inactive individuals before and after a 10-day training protocol in normoxic (21% O_2_) and normal-temperature conditions (23 °C) of the campaign number 4 within the X-Adapt: Cross-adaptation between heat and hypoxia project (Appendix A) [12]. The aim of the X-Adapt study itself was to investigate the effects of a 10-day exercise protocol on aerobic performance in young males. The X-Adapt training sessions were composed of controlled 60 min normoxic and normobaric (~1000 hPa) exercise [12] utilizing prescreened participants (graded exercise test on a cycle ergometer to determine their normoxic VO_2_max and maximal power output (W_peak_—the highest workload sustained by incremental exercise until exhaustion). In short, aerobic fitness was defined using maximal oxygen uptake (VO_2_max) values (untrained VO_2_max < 45 mL·kg^−1^·min^−1^; trained VO_2_max > 55 mL·kg^−1^·min^−1^) [13,14]. Untrained participants were also required to not participate in organized sports, while minimal cycling and walking for commuting to work were allowed. In contrast, trained participants performed endurance-type activities (running, cycling, swimming) several times per week.

The X-Adapt urine-sample collection produced by the original project outline described before [12,15,16] (i.e.,) was augmented by including two additional urine-sampling periods, extending the project outline and resulting in the extended sample collection (Figure 1 and Appendix A). As a result of these extended urine-sampling periods there was no effect on human physiology or exercise approaches utilized in the X-Adapt project. The extended sample collection included the additional three-day baseline urine samples before the actual start of the X-Adapt campaign 4 [12] and samples collected during the last three days of the 10-day exercise session. In total, the time span between the two sampling periods of extended sample collection contained 3 days testing, 1 day rest and 10 days exercise, amounting to almost 14 days of exercise [12]. This enabled us to capture the daily variability between the trained and untrained groups before the actual onset of the X-Adapt campaign 4 and to observe the actual systemic differences in response to the almost 14-day concerted exercise between the trained and untrained groups. In addition, this enabled us to perform additional comparisons between the various sections based on ^1^H-NMR urine metabolomes collected uniquely over three consecutive days (Appendix A).

As there is a lack of data and understanding on the differences between healthy trained and untrained young males and the progressive changes in human metabolomics responses coupled to introduction of exercise, we first compiled and performed a multivariate analysis of the exercise dataset [12,15,16] and hypothesized that (i) significant differences existed in the exercise dataset between trained and untrained groups; (ii) the 2-week experimental setup would enable us to detect overall change in resting urinary metabolome 3-day sequences; (iii) significant differences existed between the trained and untrained group’s urinary metabolomes despite the nonsynchronized diet of participants; (iv) the introduction of scheduled 2-week physical exercise would significantly change urine ^1^H-NMR metabolomes in the untrained group at least; (v) discriminant metabolites could be identified between the trained and untrained groups; (vi) the extended X-Adapt experiment utilized in this study provided insight into the significantly different metabolic pathways between the trained and untrained experimental variants, signifying the importance of the training history of participants for responses in human metabolomes that were also linked to the VO_2_max values (the maximal rate of oxygen consumption).

## 2. Results and Discussion

Twenty male participants 23.5 ± 2.5 years old were recruited for this study and were divided into two groups (10 participants per group) based on their physical performance (trained and untrained group). Appendix A represents their baseline characteristics. Participants in the trained group were 23 ± 2 years old, 180 ± 5 cm tall, weighed 74 ± 3 kg and had a body surface area of 1.96 ± 0.08 m^2^ and body fat of 9.2 ± 2.3%. On the other side, untrained participants were 25 ± 3 years old, 179 ± 3 cm tall, weighed 85 ± 14 kg, had a body surface area of 2.05 ± 0.17 m^2^ and body fat of 16.3 ± 4.9%. VO_2_peak, W_peak_ and W_peak_ per kg were significantly different between the untrained and trained group. The untrained group had a lower VO_2_peak (42 ± 5 mL/kg × x min in untrained and 58 ± 6 mL/kg × x min in trained group), lower W_peak_ performance (309 ± 46 W in untrained and 364 ± 35 W in trained group) and lower W_peak_ per kg (3.6 ± 0.4 W/kg in untrained and 4.9 ± 0.5 in trained group) [12,15,16,17]. 

### 2.1. Integrated Analysis of Exercise Data and the X-Adapt Urine-Sample Collection

In this study, exercise data reported before [17] and ^1^H-NMR metabolomic data obtained in this study were explored. The previously reported physiological data [12,15,16,17] relevant for metabolomic analyses within the same 3-day series of X-Adapt pre/post-testing were analyzed. Their integrated analysis in this study showed that significant multivariate differences existed between the trained and untrained groups at pretesting (Figure 1) before the onset of the 10-day 50% W_peak_ training session and after the training (PERMANOVA; F = 7.304; p(same) = 0.0001; n_permutations_ = 5000). In addition, nearly significant differences (*p* = 0.054) existed between the pre-exercise untrained and postexercise untrained groups, suggesting a larger magnitude of changes in human exercise-related characteristics than in those leading active lifestyles.

The nonmetric multidimensional scaling (nmMDS) results also showed significant groupings separating trained from untrained (Figure 2) showing that significant differences at the level of human exercise data also remained detectable after the 10-day training period. A heatmap (Figure 3 and Appendix A) of the measured exercise parameters shows large differences in measured parameters between trained and untrained groups, but also reflects significant interpersonal variability within each of the measured parameter. This suggests that although significant differences in the multivariate description of exercise states can be reported for the trained and untrained groups before and after the training sessions, the rate of change within the 10-day training at 50% W_peak_ was significantly higher for the untrained group, as reported before [12]. This observation is further supported by detailed analyses of the exercise parameters contributing most to differences between trained and untrained groups, as VO_2_max values in fact decreased 3.2% and increased for 9.2% in trained and untrained groups, respectively. This observation is in line with past observations showing that the pretraining VO_2_peak and percentage change in VO_2_peak with training were inversely correlated, showing that the rate of adaptation is largest in less physically prepared participants [18]. In addition, the integrated exercise data reported in this study showed that trained and untrained groups responded differently, as VO_2_max of the trained group could not be sustained by 50% W_peak_ training in comparison to a further increase in the untrained group in response to 50% W_peak_ training. Taken together, these results show that during the 10-day 50% W_peak_ training, the trained and untrained groups were becoming more synchronized in terms of measured exercise parameters, as also suggested before [12,15,16,17]. A two-way PERMANOVA confirmed that participant status (trained or untrained) and 50% W_peak_ training exercise (pre- or post-training) were significantly associated with the underlying multivariate exercise data (F = 13.07; F = 2.57 and p(same) = 0.0001; p(same) = 0.038), respectively), while interaction between participant status (trained or untrained) and time of training exercise (pre-or post-training) was not significant (status x exercise; F = 0.47; p(same) = 0.79), suggesting that the response of the two groups to the application of exercise was not uniform.

In contrast to results from physiological measurements, our ^1^H-NMR analyses of the X-Adapt urine-sample collection (i.e., the 3-day morning urine samples taken within the same timeframes of X-Adapt pretraining and post-training test sessions) did not identify any significant difference between any groups (PERMANOVA; *p* > 0.3; n_permutations_ = 5000) (Figure 1). This is in line with past observations that metabolomes at rest (e.g., systemic morning urine samples) cannot be indicative of physical status and capacity due to their gradual return to baseline within 24 h after exercise [19]. In addition, these results point to the potentially homogenizing short-term responses in trained and untrained individuals to the standardized pre- and post-testing conducted on three consecutive days utilized in the X-Adapt study [12]. The normoxic, temperature and hypoxic tests utilized in X-Adapt were described in detail before [12,15,16,17]. Moreover, additional in-depth tests of statistical significance between ^1^H-NMR metabolomes from trained and untrained groups on a day-to-day basis also did not produce significant differences (PERMANOVA; *p* > 0.05; n_permutations_ = 5000). These results show the lack of significant differences between the trained and untrained groups on the level of urine ^1^H-NMR metabolomes in response to the X-Adapt pretesting and post-testing trials (Figure 1). 

### 2.2. Differences in Urine ^1^H-NMR Metabolomes between the Trained and Untrained Groups: The Extended Urine-Sample Collection

In order to elucidate the potentially homogenizing responses in trained and untrained individuals to the X-Adapt training regimen the extended urine-sample collection (Figure 1) was analyzed by ^1^H-NMR. The ^1^H-NMR fingerprints of trained and untrained groups were compared to identify the existence of internaldata-structure characteristics for the two groups of participants. The results of one-way and two-way PERMANOVA showed that significant differences existed between metabolomes of trained and untrained participants (*p* < 0.01). This was also confirmed by the two-way PERMANOVA test, showing that activity (trained/untrained) was the only parameter significantly associated with the two groups (*p* = 0.0001). Regime (pre- or post-test) and the interaction between regime and activity was insignificant (*p* > 0.05). This was also confirmed by the nonsignificant change in the number of metabolites present and the sum of their concentrations in all sampled groups (Appendix A).

The Mann–Whitney test showed that significant differences in distributions between trained and untrained group existed in physical characteristics of urine such as pH, salinity, total dissolved solids (TDS) and conductivity. Salinity, conductivity and TDS were significantly higher in the untrained group than in trained, while pH was slightly more alkaline in trained (Appendix A).

Based on the nonparametric approaches described below, we used statistical methods implemented in MetaboAnalyst 5.0 [20,21,22]. According to the partial least-squares-discriminant analysis (PLSDA) of variable importance in the projection (VIP) scores, differences existed between the trained and untrained groups of participants at the level of cholate, tartrate, cadaverine, lysine and N6-acetyllisine (HMDB0000206) as the most distinguishing metabolites to differentiate the trained and untrained groups (Figure 4). The first three metabolites were all present at higher concentrations in the untrained group while concentrations of lysine and N6-acetilysine were higher in the trained group. Primary bile acid synthesis, glutathione metabolism, aminoacyl-tRNA biosynthesis and lysine degradation pathways were enriched in the untrained group (Figure 5).

In addition to the multivariate analyses, we also performed extensive machine-learning modeling using Just Add Data Bio (JADBIO) [23] to investigate the importance of metabolites and physicochemical parameters in urine samples. A total of 181,020 models were trained using extensive tuning effort. The most interpretable model was logistic ridge regression with the penalty hyperparameter lambda of 10^−4^ and an area under the curve (AUC) value of 0.748. In addition to AUC (Figure 6a); all other thresholds were also statistically significantly different from baseline. Data were preprocessed and standardized by imputation of means and removal of constants. Features were selected based on the test-budgeted statistically equivalent signature (SES) algorithm with the following hyperparameters: maxK = 2, alpha = 0.1, and budget = 3 × nvars. PCA plot (Figure 6b) shows that differentiation based on modeled data is not complete, which means that larger groups should be formed in the future. 12 metabolites and pH were selected as the most important features for distinguishing the trained from the untrained group based on urine. Appendix A lists all the important metabolites. The major metabolite selected by JADBIO was tartrate. The power of the model obtained by using only tartrate was 73.8% (with 95% CI from 69.9% to 77.6%) (Appendix A). We applied the trained model to the test portion of our data (30% of our total dataset) and achieved validation performance with an AUC of 0.647. 

Some metabolites (cholate, tartrate, methanol, N-acetylglucosamine, butanone, caprate) were selected as the top 25 metabolites using the PLSDA approach in MetaboAnalyst. Both tartrate and cholate were elevated in the untrained group, which could be related to their diet. The diet of athletes is much more constant, and the diet was not standardized in the X-Adapt project. However, the decreased tartrate levels may suggest that tartrate supplementation is needed in the trained group to reduce metabolic stress, minimize muscle damage, improve hormone receptor levels, and promote recovery after resistance exercise [24,25]. L-carnitine L-tartrate supplementation increases carbohydrate oxidation rates. Endurance athletes in particular have higher carnitine uptake in skeletal muscle [24,25]. Tartrate is a nonhuman metabolite found in grapes, wine, and as an additive in foods [26]. Increased consumption of tartrate-containing foods and beverages also lowers cardiovascular risk factors such as LDL cholesterol [27,28]. Tartrate is part of glyoxylate and dicarboxylate metabolism, which was also observed in the enrichment analysis of metabolic pathways and enriched in the untrained group. Glyoxylate and dicarboxylate metabolic pathways were observed in young patients with major depressive disorder. Improving physical activity improved patients with major depressive disorder and additionally reduced other complications of cardiovascular disease [29,30]. Inactivity in the bed rest study (e.g., PlanHab) also led to the development of psychiatric problems after one week of bed rest, showing possible associations between inactivity, metabolism and mental health problems [4,5,31,32,33,34,35,36,37,38,39,40,41,42,43,44].

Cholate, on the other hand, is one of the primary bile acids that may be involved in the development of an atrophic state in myotubes [45] and in the invasion of human colon cancer cells [46], which can be observed in less active and untrained individuals. Bile acids in general have also been associated with obesity [47], higher BMI, elevated blood glucose levels [48], liver dysfunction [49] and cardiovascular health [50]. Bile acids in urine can be used for diagnostic purposes, as it has already been shown that bile acids in urine have lower variability and higher stability than bile acids in serum [51]. Elevated cholate concentrations have also been observed in patients with gastric cancer. In our work, increased concentrations of cholate were observed in untrained individuals. A meta-analysis has previously shown that regular physical activity can prevent gastric cancer [52,53,54]. A single training run in amateur runners resulted in a significant decrease in circulating bile acids. Recent studies have also shown that bile-acid concentrations were higher in less fit women than in fit women [55,56].

Polyamines such as lysine and cadaverine, which were also detected in our study, have also been associated with the development of various diseases described by the common term “metabolic syndrome”. It has already been shown that elevated cadaverine concentrations may correlate with intestinal disease or colon and liver cancer. Cadaverine was also elevated in the untrained group and is part of the glutathione metabolism previously described in men with type 1 diabetes [57]. Metabolic syndrome develops mainly due to inactivity or lack of exercise [58,59,60]. Lysine is involved in aminoacyl-tRNA biosynthesis and was increased in trained group. Aminoacyl-tRNA biosynthesis was associated with higher physical activity, a less sedentary lifestyle and high-intensity interval training [61,62,63]. Using metabolomes in stool and serum, the same metabolic pathway was identified as altered in endurance cross-country athletes, reflecting modifications in protein synthesis [64]. 

In contrast, 2-hydroxy-3-methyl- valerate was identified only with machine learning and was decreased in the trained group, confirming that it may also be involved in affecting physical function through peroxisome proliferator-activated receptor alpha (PPAR-α) activation, which is associated with microbial metabolism and insulin sensitivity [65]. PPAR-α is a hormone-receptor transcription factor involved in energy metabolism. Untrained participants in X-Adapt are less physically active and have increased levels of 2-hydroxy-3-methyl valerate, leading to possible activation of PPAR-α, as shown in functionally impaired older adults [65,66]. N6-acetyl-L-lysine is an acetylated amino acid that is increased in the trained group and plays an important role in regulating gene transcription, cell-cycle progression, apoptosis, DNA repair and cytoskeletal organization, also decreasing chances of Alzheimer’s disease shown on rats. Physical activity has previously been shown to reduce the risk of age-related Alzheimer’s disease [67,68] and metabolic syndrome [69].

We also observed that an increased pH increased the chance of classifying participants into a trained group. A lower urine pH was associated with chronic kidney disease [70], chronic heart failure [71] and metabolic syndrome [69,72,73].

Our analyses of the same 3-day-series data on a daily basis did not produce interpretable patterns of significant differences between the daily metabolome groups of the same 3-day sampling campaign after the correction for multiple comparisons (PERMANOVA; *p* > 0.05; n_permutations_ = 5000) and were not reported. This corroborates our past observation [5] on the higher resolution of 3-day series of ^1^H-NMR metabolomes in contrast to single-day sampling. 

### 2.3. Differences between Trained and Untrained Groups before and after Synchronizing Normoxic Training Campaign: The Extended Urine-Sample Collection

Our last analysis focused on the exploration of the extended urine-sample collection between trained and untrained (Figure 1B) to identify differences in morning urine metabolomes as a result of their original lifestyle and almost 2 weeks of 1h training at 50% W_peak_ (i.e., 3-day exercise tests, 1 day rest, 10 days 1 h training at 50% W_peak;_
Figure 1A). The results of PERMANOVA (p(same) = 0.003; n_permutations_ = 5000; Appendix A) showed that in the trained group, an active lifestyle supported significantly different metabolomic fingerprints in comparison to the untrained group (Appendix A). The differences between the trained and untrained groups were no longer significant at the end of training (*p* = 0.226), while shared metabolomics features were present within each of the groups on the relation between pre- and post-training states (horizontal lines; Appendix A) as the significant differences persisted in relation to pretrained vs. post-untrained and pre-untrained vs. post-trained (diagonal lines; Appendix A). The results of this study suggest that exercise introduced changes in trained and untrained groups, making their endpoints not significantly different, and was accompanied by the concomitant decrease in the VO_2_max values (−3.2%) in trained and increase (+9.2%) in untrained groups [12,15,16,18]. 

When all eight groups of metabolomes (Figure 1B) were analyzed, it became apparent that the first introduction of controlled exercise at pre-exercise tests generated rather similar resting morning metabolomic urine makeup (i.e., short-term multivariate phenotype) in the two physiologically significantly different groups, while measurable changes within the exercise parameters (long-term multivariate phenotype (Figure 2 and Appendix A) were detected much later. Consequently, the frequency of these training bouts (i.e., life-long exercise) is in fact a crucial parameter for maintaining a healthy metabolomic phenotype and VO_2_max next to other exercise-related parameters. In contrast to WHO’s proposed 75 min to 150 min of vigorous- to moderate-intensity training, respectively, for adults per week [1,2], our study showed that a 5 times larger exercise input was effective at bringing the urine metabolomics makeup and VO_2_max values closer to the trained group, while obviously for the maintenance of an active lifestyle pursued by the trained group, much higher efforts would need to be invested. This finding is also in-line with the past observations on the difficulties in observing differences between training regimes [74], the effects of which subsided within 3 h after exercise, even in clinical populations [19]. Putting it simply, long-term exercise makes us rather similar in health, but a lack of it makes us different in disease. X-Adapt findings presented in this study on homogenizing effects of exercise are mirroring our past results from the PlanHab project on negative effects of inactivity [4,5,33,34].

To conclude, morning urine, especially as utilized in the form of 3-day sessions, has been shown to represent a good candidate biological matrix for delineation of active and inactive lifestyles in this study, detecting differences unobservable by single-day sampling. Resting morning urine metabolomes as a result of 1 h 50% W_peak_ daily activity provided a good basis for planning future recommendations for the maintenance of health in adults, irrespective of the starting fitness value. The maintenance of systemic homeostasis and the response to nutritional and environmental challenges require the coordination of multiple organs and tissues. To respond to various metabolic demands, the human body integrates and builds upon a system of interorgan communication through which one tissue can affect metabolic pathways in a distant tissue. Dysregulation of these lines of communication through lack of exercise (sedentary lifestyle) and of highly energetic diets contribute to human pathologies, including obesity, diabetes, liver disease and atherosclerosis. Increasing exercise levels in the untrained apparently has the capacity to significantly reconstitute the interorgan communication towards the levels observed in the healthy trained cohort. In addition, recent technical advances such as data-driven bioinformatics on layers of information (microbiome, proteome, metabolome) expanded our understanding of the complexity of systemic metabolic crosstalk and its underlying mechanisms [75]. 

## 3. Materials and Methods

### 3.1. Project Description

In this study, the fourth campaign of the X-Adapt: Cross-adaptation between heat and hypoxia–novel strategy for performance and work-ability enhancement in various environments project (ARRS research project J5-9350) was utilized as source of exercise data and urine samples for ^1^H-NMR metabolomics analyses (Appendix A). 

The main objective of the X-Adapt project was to determine the metabolic differences between trained and untrained individuals and the effects of 10 days of training on metabolism utilizing urinary metabolomics. 

During the prescreening procedure, participants completed a graded exercise test on a cycle ergometer to determine their normoxic (environment with normal O_2_ concentrations (e.g., 21%)) maximal-rate oxygen consumption (VO_2_max) and maximal power output (W_peak_). W_peak_ is defined as the highest workload sustained by incremental exercise until exhaustion. Aerobic fitness was defined using VO_2_max values. A VO_2_max of less than 45 mL·kg^−1^·min^−1^ or greater than 55 mL·kg^−1^·min^−1^ was considered a requirement for participation in the lower-fitness (untrained) or higher-fitness (trained) group, respectively, consistent with values reported in previous studies [13,14]. To further ensure that VO_2_max reflected participants’ true cardiorespiratory fitness levels, untrained participants were also required to not participate in organized sports. Cycling and walking for commuting to work were allowed. Accordingly, trained participants performed endurance-type activities (running, cycling, swimming) several times per week. Participants were informed that the aim of the study was to investigate the effects of a 10-day exercise protocol on aerobic performance in young males [12,15,16,17].

Twenty healthy young male volunteers were recruited to participate in the study. Inclusion criteria included males between the ages of 18 and 30, nonsmokers, and unmedicated. All participants lived near the sea and had not been exposed to altitudes > 1500 m or temperatures > 30 °C for at least 1 month before the start of the study, which took place in November and December 2018. None of the participants had a history of cardiorespiratory or hematologic disease. Participants were instructed to abstain from caffeine and alcohol consumption throughout the study. They were given detailed information about the study protocol and potential risks. 

The study consisted of three parts: pretraining exercise testing, a 10-day exercise training program, followed by the post-training exercise testing (Figure 1). 

During the pre-exercise tests, participants completed the same maximal exercise performance test on three consecutive but separate days under thermoneutral normoxic, thermoneutral hypoxic and hot normoxic conditions as described before [12,15,16,17]. The order of exercise tests was randomized and counterbalanced between participants. All tests were performed at the same time of day for a given participant (±1 h). Exercise training sessions took place in the morning hours (9:00–12:00). Participants were given a 24 h rest period before and after the 10-day training to minimize the contribution of fatigue during the exercise tests [12,15,16,17].

During the 10-day training session, all participants completed 60 min supervised cycling sessions daily for 10 days. Exercise was performed on a cycle ergometer (Daum, Electronic, Furth, Germany). During training, each participant pedalled at a preferred cadence (between 60 and 90 rpm), which they maintained throughout the experiment via visual and verbal feedback. Exercise intensity was relatively similar for all participants and was set at 50% of the W_peak_ calculated from the individual W_peak_ achieved during the preparatory graded normoxic exercise test. Participants were only informed of the time remaining until the end of the exercise session and were allowed to drink ad libitum during each exercise session. Heart rate and SpO_2_ were measured with a finger pulse oximeter (Wristox 3100 Nonin, Plymouth, MN, USA) at 5 min intervals. Ratings of perceived exertion (RPE; 6–20) was also recorded at 5 min intervals. Ambient temperature was maintained at 24 °C. The training room was well-ventilated so that normoxic and normocapnic (normal arterial carbon dioxide pressure) conditions prevailed during training. Participants completed all exercise sessions at the same time of day. No other exercise training was allowed during the study. Sessions were supervised by at least two researchers to record exercise data and ensure that all participants maintained the desired workload at all times [12,15,16,17].

After the completion of 10-day training session, postexercise tests were performed. All pre- and postexercise tests were performed in a laboratory 300 m above sea level (Ljubljana, Slovenia). Trials were performed on a cycle ergometer (Daum, Electronic, Furth, Germany) and included two phases: a 30 min steady-state workout immediately followed by incremental training to exhaustion. Before (pre) and after (post) the 10-day training protocol, participants performed three trials on three consecutive days. At normal temperature and normoxic conditions (NOR), participants breathed room air (pre: partial pressure of oxygen in the inspired air (PiO_2_) = 143.7 ± 0.8 mmHg, post: PiO_2_ = 143.4 ± 0.7 mmHg) and exercised under thermoneutral conditions (pre: Ta = 23.2 ± 0.7 °C and relative humidity (RH) = 47.2 ± 2.2%, post: Ta = 23.2 ± 0.5 °C and RH = 46.6 ± 5.9%). In the hypoxic condition (HYP), they inspired a hypoxic gas mixture (pre: PiO_2_ = 92.2 ± 1.5 mmHg, post: PiO_2_ = 93.2 ± 1.2 mmHg) and exercised in thermoneutral conditions (pre: Ta = 22.8 ± 0.5 °C and RH = 51.2 ± 1.2%, post: Ta = 22.5 ± 0.6 °C and RH = 51.5 ± 1.3%). In the hot condition (HE), the participants inspired room air (pre: PiO_2_ = 142.6 ± 1.7 mmHg, post: PiO_2_ = 142.7 ± 1.8 mmHg), but exercised in a hot environment (pre: Ta = 34.1 ± 0.9 °C, RH = 48.1 ± 4.2%, post: Ta = 34.1 ± 1.1 °C and RH = 49.8 ± 3.0%) [12,15,16,17].

### 3.2. Sample Collection

Urine samples were collected in four sessions for 3 consecutive days to form 3-day series of urine samples for all participants (Figure 1 and Appendix A): (i) 3-day baseline data of participants before the start of X-Adapt campaign, (ii) 3-day pre-exercise testing before 10-day 50% W_peak_ training, (iii) 3-day sampling of the last days of 10-day exercise; and (iv) 3-day postexercise testing after the 10-day 50% W_peak_ training. All obtained samples were frozen at −20 °C for further analysis as described before [4,5,76]. For simplicity, the X-Adapt urine-sample collection was used to denote samples collected during the X-Adapt pre-exercise and postexercise testing periods. The extended sample collection encompasses the urine samples collected before the actual onset of the campaign (baseline) and during the last three days of the 10-day 50% W_peak_ training. 

### 3.3. NMR Metabolomics

All collected samples were centrifuged (1.5 mL) at 10,000× *g* for 30 min to remove fine particles. Then, 600 µL of supernatant was mixed with 300 µL ^1^H-NMR buffer as described before [77] and stored at −25 °C until analysis. Before analysis, samples were thawed at room temperature and transferred into a 5 mm NMR tube. TSP was used as an internal standard for quantification, as described before [77].

A Bruker Avance NEO 600 MHz spectrometer equipped with a 24-sample SampleCase autosampler and a 5 mm HCN Cold probe was used for the acquisition of NMR spectra at 25 °C. The 1H NMR spectra of the samples were recorded with a spectral width of 9.0 kHz, relaxation delay of 2.0 s, 32 scans and 32 K data points. A double-pulsed field gradient spin echo (DPFGSE) pulse sequence was used for water suppression. Total correlated spectrum (TOCSY) was measured with 1H spectral widths of 7.0 kHz, 4096 complex points, a relaxation delay of 1.5 s, 32 transients and 144 time increments. An exponential and cosine-squared function were used for apodization. Zeros were filled before Fourier transform. TopSpin v. 4.0.9 software (Bruker, Billerica, MA, USA) was used for processing urine NMR spectra [4,5,76,78]. AlpsNMR R package was used for the visualization of example spectra [79]. 

### 3.4. Physicochemical Parameters of Urine Samples

Urine samples were thawed at room temperature, homogenized. Additional physical chemical parameters were recorded such as pH, conductivity, total dissolved solids and salinity using Pocket pro^+^ Multimeter 2 (Hach Company, Loveland, CO, USA). 

### 3.5. Statistical Analysis and Machine Learning

The resulting spectra were consequently analyzed using targeted quantitative metabolomics using Chenomx NMR Suite version 8.6 (Chenomx, Inc., Edmonton, AB, Canada). For the latter, all spectra were randomly ordered for spectral fitting using the ChenomX profiler and the Human Metabolome Database (https://hmdb.ca/ (accessed on 24 April 2022)) compound names were used [80]. In this study, spectral deconvolution utilizing Chenomx and HMDB was used instead of the binning approaches with extensive normalization as described before [81,82]. An ensemble approach to data analysis was utilized, employing three different approaches to asymmetric sparse matrix data analysis, establishing significant differences between tested groups as follows: nonparametric MANOVA (PERMANOVA) [83], MetaboAnalyst [20,21,22], and JADBIO [23]. Heatmap of measured physiological parameters was generated using gplots R package. Data were normalized with scale function. 

First, for PERMANOVA, each compound concentration obtained was analyzed as described before [4,11]. Box–Cox transformation was used. The significance of metabolic differences between various groups of samples was tested using 1-way and 2-way PERMANOVA, and expressed as an overlap in nonmetric multidimensional scaling (nm-MDS) trait space (using Euclidean distance measures). The stress function was used to select the dimensionality reduction, whereas Shepard’s plots were used to describe the correspondence between the target and obtained ranks. Benjamini–Hochberg significance correction for multiple comparisons was used as described before [4,5].

Second, for MetaboAnalyst, log- or cube-root transformation in connection to Mean or Pareto scaling was utilized as implemented in MetaboAnalyst, followed by supervised classification using partial least-squares-discriminant analysis (PLSDA) method and random forest (RF). Statistical power for the identification of significant differences before and after treatment was also calculated using MetaboAnalyst Statistical Power module. 

Metabolite Set Enrichment (MSEA) was used to identify biologically significant patterns between quantitative metabolome data from different groups. HMDB compound names were used to link to the KEGG database. Enrichment analysis was performed using the globaltest package implemented in MetaboAnalyst. The enrichment ratio was calculated by dividing observed hits and expected hits.

Finally, Just Add Data Bio (JADBIO), a web-based auto-machine-learning platform for analyzing potential biomarkers [23], was used. JADBIO 1.4.0 with extensive tuning effort and 6 CPU was used to model various dataset selections next to the overall 336 metabolites observed in urine samples in all groups (trained vs. untrained) by splitting the total urine metabolite data into a training set and a test set in a 70:30 ratio. The training set was used for model training and the test set was used for model evaluation.

The resulting model can be obtained as part of Appendix A and run with Java executor for the classification of novel urine samples based on ^1^H-NMR in further explorations.

## Figures and Tables

**Figure 1 metabolites-12-00473-f001:**
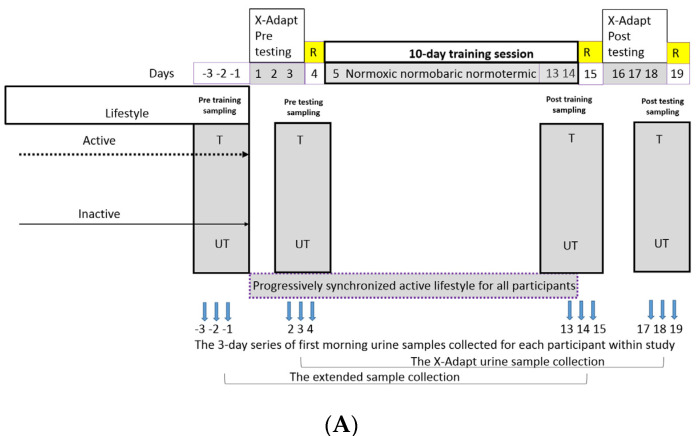
A schematic outline of the X-Adapt project with the two sampling collections designated below: the X-Adapt urine-sample collection and the extended sample collection (**A**). The extended sampling was conducted at days −3, −2, −1 day before the start of the campaign and at days 13, 14, 15 of the X-Adapt campaign 4. T—trained; UT—untrained group of participants. Blue arrows indicate sampling days within each of the four 3-day urine-sampling series. The X-Adapt urine-sample collection thus encompasses samples collected during the X-Adapt pretesting and post-testing periods. The extended sample collection encompasses the urine samples collected before the actual onset of the campaign (baseline) and during the last three days of training (days 13, 14, 15 of the campaign). For simplicity, the collection days are linked by hyphens to mark the compatible datasets. (**B**) A schematic representation of the X-Adapt urine-sample collection and the extended sample-collection groups with their respective analyses and comparisons delineated with lines. Solid and dashed lines designate significant and not significant differences between the groups. Analyses were conducted on overall group, sample collection and daily basis separately.

**Figure 2 metabolites-12-00473-f002:**
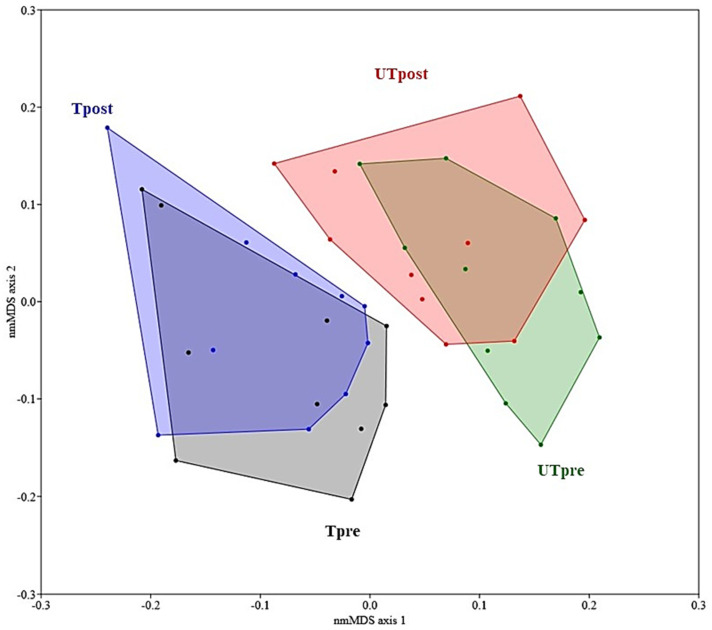
A nmMDS representation of physical parameters (*n* = 39) measured in X-Adapt project (UTpre (green)—untrained pre-exercise testing, UT-post (red)—untrained postexercise testing, Tpre (black)—trained pre-exercise testing, Tpost (blue)—trained postexercise testing). Stress value of nmMDS was 0.185. Please also see Appendix A for more details.

**Figure 3 metabolites-12-00473-f003:**
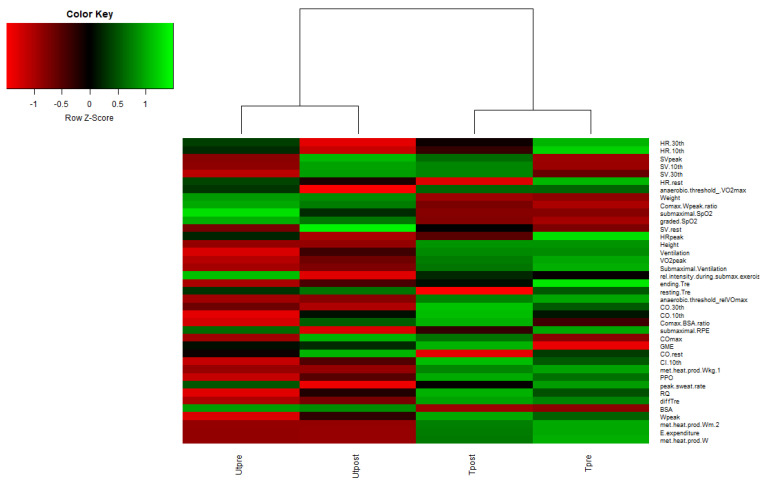
Heatmap representation of the underlying multivariate exercise physicochemical parameters measured in the X-Adapt project. Dendrogram clustering shows differences between the trained and untrained groups in these parameters before and after training. Higher-resolution heatmap can be found in electronic supplementary material. Abbreviations: body surface area (BSA), stroke volume (SV), hearth rate (HR) cardiac output (CO), rectal temperature (Tre), maximal power output (W_peak_), peak power output (PPO), gross mechanical efficiency (GME), respiratory quotient (RQ), cardiac index (CI). Please see Appendix A for the heatmap on representation per sample basis of all participants before and after exercise performance.

**Figure 4 metabolites-12-00473-f004:**
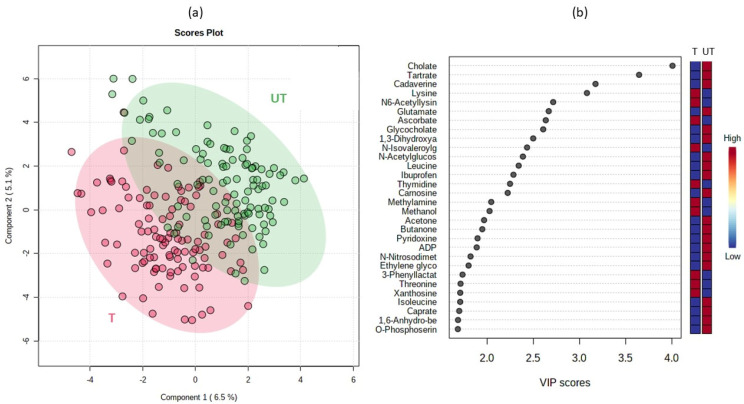
PLSDA ordination of metabolomics signatures present in the trained and untrained groups (**a**,**b**) VIP scores of the most important metabolites separating the two groups. The three-day series of urine samples of trained and untrained groups were analyzed with MetaboAnalyst. Prior to the PLSDA analysis, concentrations were transformed with Log10 normalization and scaled with Mean Centering approach. Each dot represents one sample of participant per day.

**Figure 5 metabolites-12-00473-f005:**
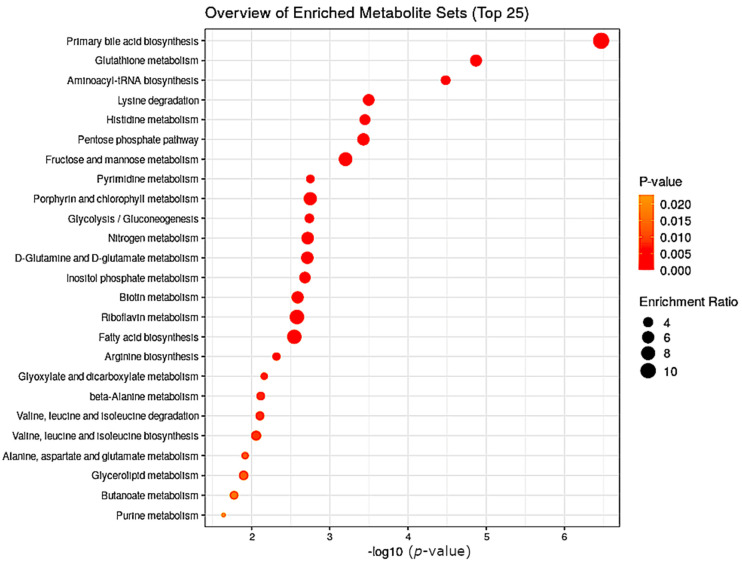
Primary bile-acid biosynthesis, glutathione metabolism and aminoacyl-tRNA biosynthesis were enriched in untrained group, based on increased levels of cholate (primary bile-acids biosynthesis), cadaverine (glutathione metabolism) and L-tryptophan and L-cysteine (aminoacyl-tRNA biosynthesis).

**Figure 6 metabolites-12-00473-f006:**
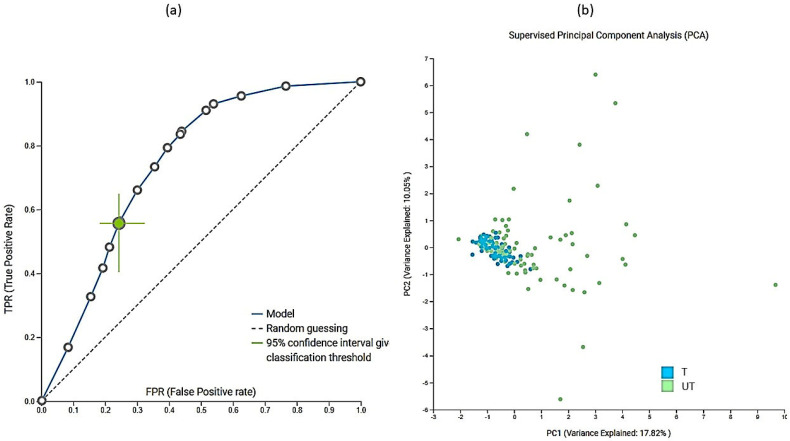
The receiver operator curve (ROC) (**a**) plot and PCA plot (**b**) of modeled data. The dimensionality reduction was performed within JADBio on a subset of the original dataset, keeping only the features included in the first signature. Features were standardized with statistical normalization ((x − µ)/σ). A total of 155 samples were included in this analysis for training the model from the entire dataset. A total of 72 samples belonged to trained and 82 samples belonged to untrained group.

## Data Availability

The data presented in this study are available in article and Appendix A.

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
