# Peer review of "Exercise and Interorgan Communication: Short-Term Exercise Training Blunts Differences in Consecutive Daily Urine 1H-NMR Metabolomic Signatures between Physically Active and Inactive Individuals"

_metabolites, 2022, doi:10.3390/metabo12060473_

Round 1

Reviewer 1 Report

The manuscript should according to the heading evaluate results from 1H NMR metabolomics data on differences between physically active and inactive individuals.  However, it describes in a very broad manner a previously done project and lists only some results without providing any real NMR data – nor spectra or tables with the corresponding shifts (not in the main text or in the supplementary data). No recommendation for publication in the present form could be provided.

Additionally, a number of explanations for the previously done experiments are written several times in the manuscript that should be avoided (e.g. 10 day training session for 10 days, introduction and project description partially overlap). An acronym description is missing, as well as explanation of some terms used  (e.g. normoxic/normocapnic). The text should be checked for a small number of typing errors.

Author Response

Response to Reviewer 1 Comments

Point1: The manuscript should according to the heading evaluate results from 1H NMR metabolomics data on differences between physically active and inactive individuals.  However, it describes in a very broad manner a previously done project and lists only some results without providing any real NMR data – nor spectra or tables with the corresponding shifts (not in the main text or in the supplementary data). No recommendation for publication in the present form could be provided.

Response 1: A: Example spectra were provided in the revised version of the accompanying Electronic Supplementary Materials. Tables containing metabolites and metadata are available through reasonable request to the corresponding author and are being prepared for submission to MetaboLights (https://www.ebi.ac.uk/metabolights/), a process that takes several weeks to finalize. This information is now part of the revised manuscript as well.

Point2: Additionally, a number of explanations for the previously done experiments are written several times in the manuscript that should be avoided (e.g. 10 day training session for 10 days, introduction and project description partially overlap). An acronym description is missing, as well as explanation of some terms used  (e.g. normoxic/normocapnic). The text should be checked for a small number of typing errors.

Response 2: Surplus explanations were omitted throughout the manuscript. Acronym description is now provided including additional explanation of some terms used, as suggested by this Reviewer. Manuscript was checked for minor typos.

Reviewer 2 Report

This is an interesting and a relevant manuscript on the downstream metabolic effects of short-term exercise employing NMR spectroscopy based metabolomics between physically active and inactive healthy young males.

The study is rigorous and should be of interest to the metabolomics and exercise physiology communities.

However the manuscript as it is written is difficult to follow and can be considerably shortened. I believe the flow and the brevity can considerable improved by a scientific writing editor. Some non-standard English spellings has been used in the manuscript which may also be edited, for example, N6-acetyllisine is more commonly called N-epsilon-Acetyl-L-lysine.

 It will be insightful to discuss the mechanistic basis of the observed metabolic changes  between the trained and untrained groups. 

Representative NMR spectra from the two groups should be shown in the supporting data file.

Author Response

Response to Reviewer 2 Comments

Point1: This is an interesting and a relevant manuscript on the downstream metabolic effects of short-term exercise employing NMR spectroscopy based metabolomics between physically active and inactive healthy young males. The study is rigorous and should be of interest to the metabolomics and exercise physiology communities.

Response 1: Thank you for your assessment of the information present in the manuscript.

Point2: However the manuscript as it is written is difficult to follow and can be considerably shortened. I believe the flow and the brevity can considerable improved by a scientific writing editor. Some non-standard English spellings has been used in the manuscript which may also be edited, for example, N6-acetyllisine is more commonly called N-epsilon-Acetyl-L-lysine.

Response 2: We did our best to screen the content of the manuscript to identify the surplus sections in order to make the manuscript as short a possible as suggested. In response to the use of »non-standard English spellings« of chemical compounds we note that the spellings are dutifully copied for consistency from the underlying Human Metabolome Database utilized with Chenomx software that were used for metabolite identification. For the sake of consistency with the HMDB database identifiable names we added human metabolome database ID to the N-epsilon-Acetyl-l-Lysin.

Point3: It will be insightful to discuss the mechanistic basis of the observed metabolic changes  between the trained and untrained groups. 

Response 3: A concluding paragraph describing and summarizing the implications on the mechanistic basis for the observed differences was added to the text to describe the impact of large muscle metabolites acting as chemical signals for other body constituents

Point4: Representative NMR spectra from the two groups should be shown in the supporting data file.

Response 4: Thank you, representative NMR spectra were prepared and are now part of the revised and extended Electronic Supplementary Material as suggested.

Reviewer 3 Report

I accepted to review this article as expert in NMR spectroscopy, given the manuscript title. In hindsight, other expertise areas are required to appropriately review this manuscript. Thus, my commentary is very limited. That said, as non-expert in research related to the medical field, I found the manuscript very well written and well structured, which allowed me to follow the authors’ line of thought throughout the manuscript. The experimental methods appear to be detailed clearly. As NMR spectroscopist, I would have preferred seeing some typical raw NMR data, at least in the electronic supplementary materials. Overall, notwithstanding my lack of expertise in medical research, the article appears to be ready for publication as is. I only found two very minor corrections. Figure 4b is a little bit cut off on the right side so that the h in “high” cannot be seen. Physicochemical is one word.

Author Response

Response to Reviewer 3 Comments

Point1: I accepted to review this article as expert in NMR spectroscopy, given the manuscript title. In hindsight, other expertise areas are required to appropriately review this manuscript. Thus, my commentary is very limited. That said, as non-expert in research related to the medical field, I found the manuscript very well written and well structured, which allowed me to follow the authors’ line of thought throughout the manuscript. The experimental methods appear to be detailed clearly. As NMR spectroscopist, I would have preferred seeing some typical raw NMR data, at least in the electronic supplementary materials. Overall, notwithstanding my lack of expertise in medical research, the article appears to be ready for publication as is. I only found two very minor corrections. Figure 4b is a little bit cut off on the right side so that the h in “high” cannot be seen. Physicochemical is one word.

Response 1: Thank you, representative NMR spectra were prepared and are now part of the revised and extended Electronic Supplementary Material. Figure 4b was modified according to the suggestion.  The term “Physicochemical« was correctly spelled.

Reviewer 4 Report

TITLE should be clearer on what “…blunts differences in systemic urine 1H-NMR metabolomic signatures…”physiologically means. What does it entail?

The study is well designed. Collection of the urine samples is performed in constant conditions and on a regular basis. Exercises are planned to make their intensity relatively similar for all participants to the study. The statistical power of the samples is adequate to reach the conclusion that daily activity provides good basis for planning future recommendations for maintenance of health in adults.

The manuscript is written in a fluid English and the concepts are clearly explained.

The references are pertinent and consistent with the subject matter.

Author Response

Response to Reviewer 4 Comments

Point1: TITLE should be clearer on what “…blunts differences in systemic urine 1H-NMR metabolomic signatures…”physiologically means. What does it entail?

 Response 1:  Thank you for this stimulating comment. We prepared and discussed several novel variants of the original title »Short-term exercise training blunts differences in systemic urine 1H-NMR metabolomic signatures between physically active and inactive individuals«  and listed some of them below with our second thoughts:

Tentative title “Short-term exercise training blunts differences in systemic urine 1H-NMR metabolomic signatures between physically active and inactive individuals making them physiologically more similar” is in our opinion quite redundant in the sense of providing explicit information that decreased differences result in increased similarity in physiological aspects.

Another tentative title “Short-term exercise training decreased differences in systemic urine 1H-NMR metabolomic signatures between physically active and inactive individuals due to higher adaptation of the inactive” in our opinion does not read well.

We also tested many more titles that were quite shorter but realized that with the shorter titles much information had to be omitted and hence the meaning of the main point was lost, in addition the title became less interesting. At the same time the extent of adaptations of human body in relation to exercise within XAdapt project are described within our manuscript, but are quite complex and we are unable to summarize them to our best knowledge in the form of a more acceptable title. Therefore we decided not to propose a different title as was suggested, but are open to the ideas of the reviewer on this matter.

Finally, the suggestion of the Reviewer5 to consider “consecutive daily urine samples“ in the title instead of “systemic” was included and is now part of novel manuscript title that now reads as “Short-term exercise training blunts differences in consecutive daily urine 1H-NMR metabolomic signatures between physically active and inactive individuals”.

A comment of Reviewer2 on more mechanistic insight was also incorporated into the novel tentative title of our revised manuscript:

Exercise and inter-organ communication: short-term exercise training blunts differences in consecutive daily urine 1H-NMR metabolomic signatures between physically active and inactive individuals

Point2: The study is well designed. Collection of the urine samples is performed in constant conditions and on a regular basis. Exercises are planned to make their intensity relatively similar for all participants to the study. The statistical power of the samples is adequate to reach the conclusion that daily activity provides good basis for planning future recommendations for maintenance of health in adults.

Response 2: Thank you for this insight. It took as quite some effort to assemble the results in their complexity into manageable form.

Reviewer 5 Report

To the authors:

  1. General comments:

The article entitled “Short-term exercise training blunts differences in systemic 2 urine 1H-NMR metabolomic signatures between physically active and inactive individuals” describes the metabolomic analysis using 1H-NMR of 10 active and 10 inactive male previous and after 10-days of training in urine collected through 3 consecutive days. The experimental design is novel and the results are interesting, however, there are major flaws in the manuscript that the authors should be addressed:

Major comments:

The experimental design is not clear, urine is collected at 4-time points: pre-trained, pre-testing, post-training, and post-testing. However, it is not clear which set of samples are taken for which comparison, the way it is written is very difficult to follow. So, I suggest redefining the groups or the concept for the reader to make it clearer.

Complementary to the previous comment, it is not clear the title as “systemic urine” concept, I would suggest adopting this concept. Consider, maybe “consecutive daily urine samples”. Additionally, in the abstract, it is specified that 3-day sessions of morning urine represent a good candidate biological matrix compared to single-day sampling, however, I did not find a comparison between single and 3-day samples. Please review.

Statistical analysis. Why was urine data not normalized? Urine is a biological matrix that must be normalized, as the rate of drinking water is different for each person. I suppose also that the trained group takes more water than the non-trained. The more frequent normalization strategy for NMR is this reference: https://doi.org/10.1021/ac051632c

Statistical analysis. Why the authors have chosen nonparametric MANOVA, instead of repeated measurements one-way ANOVA? Also, please review the multivariate statistics, this is the first time I see for metabolomics data the use of nmMDS plot, which is usually used in 16S RNA data. Why was not used PCA instead? And how the 3-consecutive urine samples were used in this plot? Also for the PLSDA, what are the quality parameters of the model and the scaling used, and the cross-validation? Finally, why was the purpose of applying machine learning in the study?, additionally to the multivariate and univariate analysis. Usually this type of approach needs a big number of samples and 10 per group is very small.

Results. Regarding metabolites, chemical shifts (ppm) with multiplicities should be included.

Materials and Methods. Raw data files should be uploaded into a metabolomics repository.

Minor comments:

Line 22. Define Wpeak in the abstract and in the text

Line 95. Why almost 14 days? This is not clear

Line 106. What does the individualized diet mean? this should be introduced more in detail in the introduction to understand a bit more the idea

Line 112. VO2max has not been defined

Lines133-137. This paragraph is not results but M&M, I was expecting to see the statistical differences between the groups by the exercise and between groups in the parameters measured.

Line 174-175. What does it mean that the interaction was not significant?

Line 190. This “in contrast”, means that the previous results were from single urine samples? And which of the 3 was taken?

Figure 6, PCA does not really separates T vs UT, and it is not mentioned in the text, please review.

Lines423-434. These are results

Figure 3. This heatmap is the average per group, but what happens using a hierarchical cluster of the samples?

Figure S2. Add in the footnote the definition of the abbreviations used in the rows

Figure S3. Add the definition, for the TDS abbreviation

Table S1, define the abbreviation used in the table and the groups used

Table S3. Add the definition, for the abbreviations

Be consistent with the names of the figures as Fig or Figures

Author Response

Response to Reviewer 5 Comments

The article entitled “Short-term exercise training blunts differences in systemic 2 urine 1H-NMR metabolomic signatures between physically active and inactive individuals” describes the metabolomic analysis using 1H-NMR of 10 active and 10 inactive male previous and after 10-days of training in urine collected through 3 consecutive days. The experimental design is novel and the results are interesting, however, there are major flaws in the manuscript that the authors should be addressed:

Major comments:

Point1: The experimental design is not clear, urine is collected at 4-time points: pre-trained, pre-testing, post-training, and post-testing. However, it is not clear which set of samples are taken for which comparison, the way it is written is very difficult to follow. So, I suggest redefining the groups or the concept for the reader to make it clearer.

Response1: The four time points used in this study were spepcifically designated at the beginning of the manuscript in the form of a figure. Within this figure the notations of the timepoints and the specific comparisons were designated. In addition, the comparisons made on group, time point or daily basis were used in the text. To make these points clearer we amended the manuscript and rewritten some of the portions of the materials and methods section next additions into Results and Discussio nsection to specifically provide more information on the notation of the four time points in this study.

Point2: Complementary to the previous comment, it is not clear the title as “systemic urine” concept, I would suggest adopting this concept. Consider, maybe “consecutive daily urine samples”. Additionally, in the abstract, it is specified that 3-day sessions of morning urine represent a good candidate biological matrix compared to single-day sampling, however, I did not find a comparison between single and 3-day samples. Please review.

Response 2: Thank you. The title was amended as suggested and now reads as following: Exercise and inter-organ communication: Short-term exercise training blunts differences in consecutive daily urine 1H-NMR metabolomic signatures between physically active and inactive individuals.

The comparison between the single day samples of the same groups and three-day series was described in the submitted manuscript in the 2.1 section of Results and discussion. This information was expanded in the revised manuscript as requested.

Point3: Statistical analysis. Why was urine data not normalized? Urine is a biological matrix that must be normalized, as the rate of drinking water is different for each person. I suppose also that the trained group takes more water than the non-trained. The more frequent normalization strategy for NMR is this reference: https://doi.org/10.1021/ac051632c

Response 3: Thank you for this stimulating comment. In this study, spectral deconvolution was used to identify and quantify compounds in individual NMR spectra and the resulting compound IDs and concentrations from multiple spectra are compiled to create a data matrix for multivariate statistical analysis utilizing software tools for NMR spectral deconvolution such as the Chenomx NMR Suite and HMDB database as described before (Murovec 2018; Sket, 2018; 2020; Deutsch 2021,22).

In this sense our analysis was not  related to the second approach to NMR data analysis that  initially focuses on steps to align multiple NMR spectra, to scale or normalize the aligned spectra, and then to identify interesting spectral regions (e.g. binning) or peaks that differentiate cases from controls that  performs compound identification or quantification only after the most interesting peaks have been identified. The suggested normalization procedure falls within this second routine.

Both approaches are valid in NMR data analysis of biological fluids as was described recently in review by Emwas et al. (https://doi.org/10.1007/s11306-018-1321-4)

All methods were described in materials and methods in section Statistical analysis and machine learning. In short - Data was processed with Chenomx software and concentration of particular metabolites were obtained. Matrix with metabolites in micromolar concentration were generated and then merged with metadata. After that we used several different methods for standardization and transformation. Box-cox method was used for PERMANOVA. Log or cube root transformation next to Mean or Paretto scalling was used for analysis within the MetaboAnalyst to normalized data as much as possible. Additionally, data was also standardize automatically when building the most appropriate model within the automatic process of machine learnin in all of the 181020 trained models.

Point4: Statistical analysis. Why the authors have chosen nonparametric MANOVA, instead of repeated measurements one-way ANOVA? Also, please review the multivariate statistics, this is the first time I see for metabolomics data the use of nmMDS plot, which is usually used in 16S RNA data. Why was not used PCA instead? And how the 3-consecutive urine samples were used in this plot? Also for the PLSDA, what are the quality parameters of the model and the scaling used, and the cross-validation? Finally, why was the purpose of applying machine learning in the study?, additionally to the multivariate and univariate analysis. Usually this type of approach needs a big number of samples and 10 per group is very small.

Response 4: MANOVA was selected as the preferred approach as the groups under analysis are not independent, have unequal variance and the data are not normally distributed.

The approach of nmMDS is more robust to violations of statistical assumptions and was utilized to map the physiological data in Figure 2 and urine chemical characteristics in Figure S2. PCA of urine 1H-NMR data was presented in Figure 6 of the submitted manuscript.

In addition, several approaches described in Materials and methods section, including various modes of data normalization, were utilized in analysis of this 1H-NMR dataset.

Machine learning was used with intent to identify most important metabolites separating the trained and untrained groups based on a complete set of 239 samples in total.

Point5: Results. Regarding metabolites, chemical shifts (ppm) with multiplicities should be included.

Response 5: In this study, spectral deconvolution was used to identify and quantify compounds in individual NMR spectra and the resulting compound IDs and concentrations from multiple spectra are compiled to create a data matrix for multivariate statistical analysis utilizing software tools for NMR spectral deconvolution such as the Chenomx NMR Suite and HMDB database as described before (Murovec 2018; Sket, 2018; 2020; Deutsch 2021,22).

In this sense our analysis was not  related to the second approach to NMR data analysis that  initially focuses on steps to align multiple NMR spectra, to scale or normalize the aligned spectra, and then to identify interesting spectral regions (e.g. binning) or peaks that differentiate cases from controls that  performs compound identification or quantification only after the most interesting peaks have been identified. The suggested normalization procedure falls within this second routine.

Both approaches are valid in NMR data analysis of biological fluids as was described recently in review by Emwas et al. (https://doi.org/10.1007/s11306-018-1321-4)

Point6: Materials and Methods. Raw data files should be uploaded into a metabolomics repository.

 Response 6: Slovenian Metabolomic Database is in preparation and the data is projected to be stored within this infrastructure. We also commenced the submission of the data into MetaboLights (https://www.ebi.ac.uk/metabolights/) database, however the process of finalizing submission takes several weeks. In the mean time the raw data files are available by request to the authors as described in the submitted manuscript.

Minor comments:

Point7: Line 22. Define Wpeak in the abstract and in the text

Response 7: Wpeak was defined in materials and methods section, in abstract and in text.

Point8: Line 95. Why almost 14 days? This is not clear

Response 8:  In addition to the explanation provided within the submitted manuscript we augmented the revised version with additional explanations where needed that : In total, the time span between the two sampling periods of extended sample collection contained 3 day testing, 1 day rest, 10 day exercise, amounting to almost 14 days of exercise [12].

Point9: Line 106. What does the individualized diet mean? this should be introduced more in detail in the introduction to understand a bit more the idea

Response 8: The diet itself was not standardized, which basically means that the participants lived their own lives as always. "Individualized" was therefore changed to "non-synchronized" in hope of providing a better description as suggested.

Point10: Line 112. VO2max has not been defined

Response 9: VO2max was defined in the Materials and methods section and in the manuscript body text.

Point11: Lines133-137. This paragraph is not results but M&M, I was expecting to see the statistical differences between the groups by the exercise and between groups in the parameters measured.

Response 11: This part was removed as suggested.

Point12: Line 174-175. What does it mean that the interaction was not significant?

Response 12: Thank you, manuscript section was augmented and additional explanation was provided in the text.

Point13: Line 190. This “in contrast”, means that the previous results were from single urine samples? And which of the 3 was taken?

Response 13: We rephrased this section to make clear that “In contrast, …” was related to physiological measurements, while the presentation of results was dealing with 1H-NMR results of X-Adapt sample collection.

In addition, information was provided where relevant in the revised manuscript on the results of three-day series vs single day results (where each of the three days was used in analysis as a separate datapoint). The significance and resulting distance matrices were also explored but this constitutes an additional information layer in already complex metabolomics story therefore extensive descriptions were omitted.

Point14: Figure 6, PCA does not really separates T vs UT, and it is not mentioned in the text, please review.

Response 14: Thank you for that comment. It is true that PCA does not separate T vs UT, however, the underlying classification model was still successful in the classification of test samples at comparable metric to other such models in literature.

We added the description of the PCA plot in the main text.

Point15: Lines423-434. These are results

Response 15: This part was moved to results and discussion section as suggested.

Point16: Figure 3. This heatmap is the average per group, but what happens using a hierarchical cluster of the samples?

Response 16: We added heatmap with performed hierarchical clustering into ESM (Figure S2) as suggested.

Point17: Figure S2. Add in the footnote the definition of the abbreviations used in the rows

Response 17: Abbreviations were added.

Point18: Figure S3. Add the definition, for the TDS abbreviation

Response 18: Definition was added.

Point19: Table S1, define the abbreviation used in the table and the groups used

Response 19: Abbreviations were added.

Point20: Table S3. Add the definition, for the abbreviations

Response 20: Abbreviations were added.

Point21: Be consistent with the names of the figures as Fig or Figures

Response 21: “Fig” changed to “Figure”.

Round 2

Reviewer 1 Report

-

Author Response

Response 1: No specific point was raised that would need to be addressed.

Reviewer 5 Report

To the authors:

1.      General comments:

I would like to thank the authors for the great effort towards the edition of the manuscript. I consider that the work has improved. I only have a few minor comments:

2.      Comments:

Lines 601,603. Unify using PERMANOVA or npMANOVA

Figure 4. It is not clear if for the PLSDA the 3-consecutive urines are plotted, and what was the scaling of the data used.

Figure 3. The heatmap contains multiple abbreviations that have not been defined such as BSA, SV, PPO, and so on. Please include them in the footnote

Figure 6. please include the scaling of the data for the PCA model and indicate which samples and their number per group used.

Author Response

Response to Reviewer 5 Comments

Comments and Suggestions for Authors

To the authors:

Point1: General comments:

I would like to thank the authors for the great effort towards the edition of the manuscript. I consider that the work has improved. I only have a few minor comments:

Response 1: Thank you for this comment.

  1. Comments:

Point2: Lines 601,603. Unify using PERMANOVA or npMANOVA

Response 1: Corrected.

Point3: Figure 4. It is not clear if for the PLSDA the 3-consecutive urines are plotted, and what was the scaling of the data used.

Response 1: Thank you. Additional information that the 3-consecutive urines were plotted was entered into Figure caption.

Point4: Figure 3. The heatmap contains multiple abbreviations that have not been defined such as BSA, SV, PPO, and so on. Please include them in the footnote

Response 1: Thank you. Abbreviations were defined in figure captions.

Point5: Figure 6. please include the scaling of the data for the PCA model and indicate which samples and their number per group used.

Response 1: Additional information was presented in the figure captions as requested.
